# Comparison of a New ^68^Ga-Radiolabelled PET Imaging Agent sCD146 and RGD Peptide for In Vivo Evaluation of Angiogenesis in Mouse Model of Myocardial Infarction

**DOI:** 10.3390/cells10092305

**Published:** 2021-09-03

**Authors:** Anaïs Moyon, Philippe Garrigue, Samantha Fernandez, Fabien Hubert, Laure Balasse, Pauline Brige, Guillaume Hache, Vincent Nail, Marcel Blot-Chabaud, Françoise Dignat-George, Francesca Rochais, Benjamin Guillet

**Affiliations:** 1Pharmacological Faculty, Aix Marseille University, INSERM 1263, INRAE 1260, C2VN, 13385 Marseille, France; Philippe.garrigue@univ-amu.fr (P.G.); guillaume.hache@univ-amu.fr (G.H.); Vincent.nail@ap-hm.fr (V.N.); marcel.blot-chabaud@univ-amu.fr (M.B.-C.); Francoise.dignat-george@univ-amu.fr (F.D.-G.); benjamin.guillet@univ-amu.fr (B.G.); 2Medical Faculty, Aix-Marseille University, CNRS 2012, CERIMED, 13385 Marseille, France; samantha.fernandez@univ-amu.fr (S.F.); laure.balasse@univ-amu.fr (L.B.); pauline.brige@univ-amu.fr (P.B.); 3APHM, Service de Radiopharmacie, 13005 Marseille, France; 4Medical Faculty, Aix Marseille University, INSERM, MMG, U 1251, 13385 Marseille, France; Fabien.hubert@univ-amu.fr (F.H.); francesca.rochais@univ-amu.fr (F.R.); 5Medical Faculty, Aix-Marseille University, UR4264, LIIE, 13385 Marseille, France; 6APHM, Service d’Hématologie, Hôpital Conception, 13005 Marseille, France

**Keywords:** angiomotin, myocardial infarction, sCD146, gallium, angiogenesis

## Abstract

Ischemic vascular diseases are associated with elevated tissue expression of angiomotin (AMOT), a promising molecular target for PET imaging. On that basis, we developed an AMOT-targeting radiotracer, ^68^Ga-sCD146 and performed the first in vivo evaluation on a myocardial infarction mice model and then, compared AMOT expression and α_v_β_3_-integrin expression with ^68^Ga-sCD146 and ^68^Ga-RGD_2_ imaging. After myocardial infarction (MI) induced by permanent ligation of the left anterior descending coronary artery, myocardial perfusion was evaluated by Doppler ultrasound and by ^18^F-FDG PET imaging. ^68^Ga-sCD146 and ^68^Ga-RGD_2_ PET imaging were performed. In myocardial infarction model, heart-to-muscle ratio of ^68^Ga-sCD146 imaging showed a significantly higher radiotracer uptake in the infarcted area of MI animals than in sham (* *p* = 0.04). Interestingly, we also observed significant correlations between ^68^Ga-sCD146 imaging and delayed residual perfusion assessed by ^18^F-FDG (* *p* = 0.04), with lowest tissue fibrosis assessed by histological staining (* *p* = 0.04) and with functional recovery assessed by ultrasound imaging (** *p* = 0.01). ^68^Ga-sCD146 demonstrated an increase in AMOT expression after MI. Altogether, significant correlations of early post-ischemic ^68^Ga-sCD146 uptake with late heart perfusion, lower tissue fibrosis and better functional recovery, make ^68^Ga-sCD146 a promising radiotracer for tissue angiogenesis assessment after MI.

## 1. Introduction

Myocardial infarction (MI) is a complex disease underlying different pathophysiological processes and issues. Early reperfusion therapies such as primary coronary intervention decrease mortality. Yet, functional recovery is often limited due to residual microcirculation dysfunction [1]. Boosted angiogenesis after acute MI is associated with favorable outcome in animal models as evidenced by better preserved heart function [2]. Post-MI angiogenesis processes are still subject to intensive research and many efforts are still committed to develop innovative pro-angiogenic therapeutic strategies based on pharmacological agents, cells [3], extracellular vesicles [4] or secretome [5]. In parallel with these developments, non-invasive clinical evaluation of the angiogenic process is critical for post-ischemic risk stratification, therapeutic eligibility, follow-up and management.

A non-invasive method assessing tissue angiogenic status could therefore offer a refined prognosis real-time monitoring of treatment efficacy. Positron emission tomography coupled with computed tomography (PET/CT) may represent an efficient and sensitive imaging modality to quantify angiogenesis, but the ideal molecular target and its associated radiotracer remain to be identified, developed and validated. Indeed, almost two decades after the first developments, PET imaging agents targeting integrins or molecules of the VEGF pathway still have not reached registration [6]. Several clinical studies in patients with MI and stroke showed an increased PET signal in the ischemic region in direct correlation with the phase and severity of the disease. However, these positive results must be confirmed in a larger number of patients in order to refine the prognostic power of these tracers and their use in the evaluation of pro-angiogenic therapies [7,8,9,10]. Efforts are needed to find innovative and relevant molecular targets.

Among them, we recently reported that angiomotin (AMOT) imaging was of interest for PET imaging of angiogenesis and tissue regeneration [11]. AMOT involvement in angiogenesis is widely described both in ischemic pathology [11], in post-hypoxia adaptation [12], in chronic intermittent hypoxia with re-oxygenation [13] and also during tumour vasculature growth promotion [14,15].

AMOT was first identified as an angiostatin-binding protein involved in the regulation of endothelial cell polarization, migration, proliferation and angiogenesis [16,17,18]. More recently, the soluble cluster of differentiation 146 (sCD146) was described as an endogenous AMOT ligand promoting angiogenic effects on endothelial populations [19]. sCD146 presented angiogenic properties in vitro and in vivo in hindlimb ischemia models [20] and tumour-bearing mouse models [21]. Furthermore, numerous reports highlighted that serum levels of sCD146 could be considered as a novel biomarker of heart failure patients [22,23].

Based on these observations, we recently developed an AMOT-targeting radiotracer, ^68^Ga-sCD146 and proved its value as an angiogenesis PET imaging agent and as an early predictive tool to predict delayed revascularization in mouse hindlimb ischemia model [11]. The aim of this study was therefore to evaluate AMOT PET imaging using ^68^Ga-sCD146 for post-ischemic angiogenesis evaluation and its predictive value for myocardial tissue recovery.

## 2. Materials and Methods

### 2.1. Animals

All procedures using animals were approved by the Institution’s Animal Care and Use Committee (Project #2017040416045642 CE14 Aix-Marseille University) and were conducted according to the EU Directive 2010/63. Ten-week-old male CD1 mice (Janvier Labs) were housed in enriched cages placed in a temperature-and hygrometry-controlled room with daily monitoring and fed with water and commercial diet ad libitum.

### 2.2. Mouse Model of Myocardial Infarction

Mice were anesthetized with a mixture of ketamine (100 mg/kg) and xylazine (10 mg/kg) via intraperitoneal injection and following endotracheal intubation, were artificially ventilated. If necessary 1–2% isoflurane was added as maintenance anesthetic. For analgesia, buprenorphine. (0.1 mg/kg) was injected subcutaneously 30 min prior surgery. Following skin incision, lateral thoracotomy at the fourth intercostal space was performed by blunt dissection of the intercostal muscles. Under stereomicroscope control, the left anterior descending coronary artery was visualized and ligated (with 8.0 non-absorbable silk suture) 2.0 mm below the left atrium, just above the bifurcation of the left diagonal arteries. Effective ligation of the coronary artery was confirmed by whitening of the LV affected region below the ligation site. The thoracic wall and skin incisions were then sutured with 6.0 non-absorbable and 4.0 absorbable silk sutures, respectively. Mice were then warmed for several minutes until recovery.

### 2.3. Ultrasound Imaging

In vivo heart structure and function of CD1 mice were evaluated using a high-frequency scanner (Vevo2100 VisualSonics, Bothell, WA, USA). Briefly, mice were anesthetized with 1–2% isoflurane inhalation and placed on a heated platform to maintain temperature during the analysis. Two-dimensional imaging was recorded with a 22–55 MHz transducer (MS550D) to capture long- and short-axis projections with guided M-Mode and B-Mode. Left parasternal long axis and left short axis view in M-mode. The left ventricular ejection fraction (VEF) is expressed as % and calculated by dividing the systolic ejection volume (TDV) and the telesystolic volume STV), commonly marked ESV) by the end-diastolic volume (TDV).

### 2.4. Radiochemistry

^68^Ga-sCD146 radiosynthesis: preparation of NODAGA-sCD146, its subsequent radiolabelling, the evaluation of radiolabelling stability and in vivo biodistribution were performed as previously described [6].

^68^Ga-RGD_2_ radiosynthesis: gallium-68 chloride (^68^GaCl_3_, 200.7 ± 41 MBq/500 μL) was buffered with a fresh 2M ammonium acetate solution (pH 7.4) and 10 μg of RGD_2_ precursor (catalogue number 9804, purchased at ABX, Radeberg, Germany) was added. The reaction mixture was stirred at room temperature for 15 min. Radiochemical purity was determined by radio-thin-layer chromatography (TLC) using a miniGITA radio-TLC scanner detector (Raytest, straubenhardt, Germany) (solid phase: ITLC-SG; eluents: 1:1 [v/v] mixture of 1 M aqueous ammonium acetate solution and methanol; R_f_ [free ^68^Ga]/[^68^Ga-RGD_2_]; eluents 2: Sodium citrate 0.1M Ph = 5; R_f_ [^68^Ga-RGD_2_]/[free ^68^Ga]).

^18^F-Fluorodesoxyglucose (^18^F-FDG) was purchased as a ready-to-use radiopharmaceutical (Gluscan, Advanced Accelerator Applications, Marseille, France).

### 2.5. MicroPET/CT Imaging

Procedures for microPET/CT imaging experiments are summarized in Figure 1. MicroPET/CT acquisitions were performed on a NanoscanPET/CT camera (Mediso, Budapest, Hungary). For each PET tracer, radioactivity was injected intravenously in the retro-orbital sinus. Mice were maintained under 2% isoflurane anesthesia during acquisition. Static microPET imaging was performed 1 h after each radiotracer injection, during 20 min.

Glucose metabolism was evaluated by ^18^F-FDG imaging at early and later time after myocardial infarction. Indeed, while a normal ^18^F-FDG uptake is characteristic of viable myocardium, a low uptake of ^18^F-FDG indicates a hibernating myocardium and the absence of ^18^F-FDG uptake indicates scar tissue.

Seven myocardial infarction mice and 4 sham mice, fasted for 4 h, were IV injected with 10 ± 3.5 MBq/50 μL of ^18^F-FDG on days 16 and 30 post-ischemia. MicroPET images were acquired 1 h after injection.

Seven myocardial infarction mice and 4 sham mice were IV injected with 5.0 ± 2.4 MBq/100 μL of ^68^Ga-sCD146 on days 15 and 22 post-MI. MicroPET images were acquired 1 h after injection.

Seven myocardial infarction mice and 4 sham mice were IV injected with 5.0 ± 3.2 MBq/100 μL of ^68^Ga-RGD_2_ on days 14 and 23 post-MI. MicroPET images were acquired 1 h after injection.

Semi-quantitative region-of-interest (ROI) analysis of the PET signal was performed on attenuation- and decay-corrected PET images using InterviewFusion software (Mediso) and tissue uptake values were expressed as a mean percentage of the injected dose per gram of tissue (%ID/g) ±SD for ^18^F-FDG and as a mean heart-to-muscle signal ratio (H/M_ischemic_) ± SD for ^68^Ga-sCD146 and ^68^Ga-RGD_2_. This imaging method only enables semi-quantitative analysis of the PET signal as neither dynamic imaging nor kinetic modeling study was carried out for true quantitative analysis.

### 2.6. Histological Sirius Red Staining

Forty-three days after myocardial infarction, hearts were collected and fixed in 4% PFA for 3 h and extensively washed in 1X PBS. Paraffin embedding was performed following dehydration using a graded ethanol series (50, 70, 90 and 100%), two xylene washes and three paraffin washes (Paraplast X-tra, Sigma P3808, Saint-Louis, MS, USA). Serial 13 μm sections were obtained and mounted on polylysine-treated slides. After dewaxing (xylene, 2 times) and rehydration in an ethanol series (100, 90, 70, 50% and H_2_O), paraffin sections were incubated in a 0.1% Sirius Red solution dissolved in saturated aqueous solution of picric acid for 1 h at room temperature. Subsequently, sections were washed 3 times in acidified water (0.5% acetic acid), dehydrated in ascending concentrations of ethanol (70%, 90% and 100%) and cleared in xylene. Sections were mounted in resinous medium (Entellan, Merck, Saint-Louis, MS, USA). Collagen and non-collagen components were red- and orange-stained, respectively.

### 2.7. Statistical Analysis

Biodistribution data were analyzed using Prism software v9 (GraphPad, San Diego, CA, USA). Data were expressed as mean values ± SD. Earth to muscle ratio differences were analyzed using to Mann–Whitney test. Differences between sham and infarct hearts evaluated by immunohistochemistry and by ultrasound imaging were analyzed using parametric unpaired Student t-test. Gaussian distribution was assumed by a Shapiro–Wilk normality test. Analysis of correlation was realized with a Pearson two-tailed test. Differences were considered statistically significant when *p* < 0.05.

## 3. Results

### 3.1. Evaluation of the Myocardial Infarction Defect

^18^F-FDG microPET imaging clearly highlighted myocardial infarction defects at day 16 and tissue recovery day 30 post-surgery (Figure 2A). Semi-quantitative analysis of ^18^F-FDG uptake on ischemic mice showed a significant decrease in Apex area compared to control area (Mean_apex_ = 5.19 ± 4.10%ID/g, Mean_control_ = 30.7 ± 7.25%ID/g, *n* = 7; *p* = 0.0003 at day 16) and (Mean_apex_ = 3.84 ± 1.81%ID/g, Mean_control_ = 15.9 ± 2.81%ID/g, *n* = 7; *p* = 0.0003 at day 30). In contrast, no difference was observed in sham hearts between apex and control aera (Mean_apex_ = 24.8 ± 11.3%ID/g, Mean_control_ = 25.4 ± 12.9%ID/g, *n* = 4; *p* = 0.443 at day 16) and (Mean_apex_,14.3 ± 1.54%ID/g, Mean_control_ 16.2 ± 2.20%ID/g, *n* = 4; *p* = 0.885 at day 30) (Figure 2B).

Following surgery, cardiac function was evaluated using ultrasound imaging (Figure 3A). Our results showed that myocardial infarction led to a significant decrease of the ventricular ejection fraction (VEF) compared to sham mice (VEF_ischemic_ = 24.8 ± 8.84%, *n* = 7, VEF_sham_ = 46.2 ± 14.5%, *n* = 4; *p* = 0.009 at day 22 post-surgery) and (VEF_ischemic_ = 31.7 ± 11.4%, *n* = 7, VEF_sham_ = 46.2 ± 8.08%, *n* = 4; *p* = 0.04 at day 43 post-surgery) (Figure 3B).

At the end of the experiment (day 43 post-surgery), myocardial fibrosis was evaluated on sham and infarcted hearts using histological Sirius Red staining analysis (Figure 3C). Semi-quantitative analysis revealed that myocardial infarction resulted in a significant increase of myocardial fibrosis in ischemic hearts compared to sham hearts (Mean_ischemic_ = 10.79 ± 4.52%, *n* = 7, Mean_sham_ = 0.88 ± 0.07%, *n* = 4; *p* = 0.0061) at day 43 post-surgery (Figure 3D).

### 3.2. ^68^Ga-sCD146 Showed an Early and Sustained Increase of AMOT Expression whereas ^68^Ga-RGD_2_ Showed No Significant Increase in Integrin Expression at Day 15 following Myocardial Infarction

Semi-quantitative analysis of ^68^Ga-sCD146 microPET signal expressed as heart-to-muscle ratio, showed an significant increase in ischemic hearts at day 15 compared to sham hearts (H/M_ischemic_ = 3.23 ± 0.82, *n* = 7, H/M_sham_ = 2.03 ± 0.25, *n* = 4; *p* = 0.04) and no significant increase was showed at day 22 post-surgery H/M_ischemic_ 2.52 ± 0.94, *n* = 7; H/M_sham_ = 2.02 ± 0.56, *n* = 4; *p* = 0.30). Semi-quantitative analysis of ^68^Ga-RGD_2_ microPET signal expressed as heart-to-muscle ratio, showed no significant uptake increase on day 14 post-surgery (H/M_ischemic_= 2.92 ± 1.48, *n* = 7; H/M_sham_ = 1.26 ± 0.13, *n* = 4, *p* = 0.15) and a significant increase in ischemic hearts compared to sham hearts (H/M_ischemic_ = 0.96 ± 0.18, *n* = 7, H/M_sham_ = 0.65 ± 0.12, *n* = 4; *p* = 0.02) at day 23 post-surgery. Semi-quantitative intra-animal comparison of both tracers, ^68^Ga-sCD146 and ^68^Ga-RGD_2_ at early times, respectivly D14 and D15 after MI surgery. No significant difference was observed between ^68^Ga-sCD146 and ^68^Ga-RGD_2_ PET uptake in the infarcted heart at this time (*n* = 7; *p* = 0.93). Semi-quantitative intra-animal comparison was also realized at a later time, respectively, D22 and D23 for ^68^Ga-sCD146 and ^68^Ga-RGD_2_. Significant differences was observed between both tracers ^68^Ga-sCD146 and ^68^Ga-RGD_2_ PET uptake in the infarcted heart at this later time (*n* = 7; *p* = 0.01) (Figure 4).

### 3.3. Early ^68^Ga-sCD146 PET Signal Intensity on Day 15 Correlated with Delayed Residual Myocardial Perfusion

A significant positive correlation was observed between individual residual perfusion recovery evaluated by ^18^F-FDG PET signal expressed as heart-to-muscle ratio on day 30 and ^68^Ga-sCD146 PET signal intensity expressed as heart-to-muscle ratio (Figure 5(Aa)) on day 15 post-ischemia (Pearson R^2^ = 0.70; *p* = 0.04; *n* = 7). In contrast, no significant correlation was depicted between individual residual perfusion recovery evaluated by ^18^F-FDG PET signal expressed as heart-to-muscle ratio on day 30 and ^68^Ga-RGD_2_ PET signal intensity (Figure 5(Ab)) on day 14 post-ischemia (Pearson R^2^ = 0.21; *p* = 0.64; *n* = 7).

### 3.4. Early ^68^Ga-sCD146 PET Signal Intensity Was Inversely Correlated with Delayed Myocardial Fibrosis

Our results revealed a significant negative correlation between individual fibrosis quantification evaluated by histological Sirius red staining and expressed as percentage of fibrosis tissue on left ventricle on day 43 post-ischemia and ^68^Ga-sCD146 PET signal intensity expressed as heart to muscles ratio (Figure 5(Ba)) on day 15 post-ischemia (Pearson R^2^ = −0.77; *p* = 0.04; *n* = 7). In contrast, no significant correlation was detected between individual fibrosis quantification 43 days post-myocardial infarction and ^68^Ga-RGD_2_ PET signal intensity expressed as heart to muscle ratio (Figure 5(Bb)) on day 15 post-myocardial infarction (Pearson R^2^ = −0.68; *p* = 0.06; *n* = 7).

### 3.5. Early ^68^Ga-sCD146 PET Signal Correlated with Myocardial Functional Recovery

Most notably, a significant correlation was identified between individual heart functional recovery, explored using ultrasound imaging and expressed as a ratio between the VEF measured 43 and 22 days post-myocardial infarction and ^68^Ga-sCD146 PET signal intensity expressed as heart-to-muscle ratio (Figure 5(Ca)) on day 15 post-myocardial infarction (Pearson R^2^ = 0.68; *p* = 0.01; *n* = 12).

A significant correlation between individual functional recovery and ^68^Ga-RGD_2_ PET signal intensity was also observed 15 days post-myocardial infarction (Pearson R^2^ = 0.59; *p* = 0.04; *n* = 12) (Figure 5(Cb)).

All these correlations were also performed for ^68^Ga-sCD146 and ^68^Ga-RGD_2_ late times, respectively D22 and D23 post IM. The results of these correlations are shown in supplemental Appendix A. We have also detailed the semi-quantitative results of each imaging method in Appendix A presented in additional data.

### 3.6. Early ^18^F-FDG PET Signal Intensity Was Inversely Correlated with Delayed Myocardial Fibrosis and Wasn’t Correlated with Myocardial Functional Recovery

A significant negative correlation was identified between individual fibrosis quantification expressed as percentage of fibrotic tissue on left ventricle on day 43 post-ischemia and ^18^F-FDG PET signal intensity expressed as heart-to-muscle ratio (Figure 5(Da)) on day 16 post-ischemia (Pearson R^2^ = −0.83; *p* = 0.024; *n* = 10). In contrast, no significant correlation was detected between individual heart functional recovery explored using ultrasound imaging and expressed as a ratio between the VEF measured 43 and 22 days post-myocardial infarction and ^18^F-FDG PET signal intensity expressed as heart-to-muscle ratio (Figure 5(Db)) on day 16 post-ischemia (Pearson R^2^ = −0.39; *p* = 0.25; *n* = 10).

## 4. Discussion

In this study, we demonstrated for the first time using microPET imaging an increase in AMOT expression after MI. We additionally observed that ^68^Ga-sCD146 microPET imaging on day 15 post-ischemia was positively correlated with delayed residual myocardial perfusion, myocardial functional recovery and negatively correlated with delayed myocardial fibrosis. According to recent reports describing AMOT involvement in angiogenic processes [24], these results indicate that ^68^Ga-sCD146 may be considered as a potentially sensitive tool to early predict recovery of recent myocardial injury.

MI affects millions of patients worldwide with increasing prevalence and is responsible for millions of deaths. MI is mainly due to rupture or erosion of vulnerable atherosclerotic plaque, inducing coronary artery occlusion and cell death in ischemic territory [25]. Despite advances in pharmacological and interventional therapies, only 10% of all MI patients survive with reduced left ventricular function, resulting in the development of adverse LV remodeling and heart failure [26,27,28]. Research and development of innovative therapies remain a major challenge for such high-risk patients. Wound healing after MI involves a robust angiogenic response as a key of therapeutics approach. The post-MI angiogenic mechanisms have been poorly described up to now but several studies showed a strong involvement of pre-existing ECs in the capillary formation in infarct region [29,30]. Endothelial cells (EC) emerged as a major therapeutic target for promoting angiogenesis and tissue regeneration after MI [2]. AMOT has been shown to promote EC migration and tube formation during angiogenesis [18]. AMOT is expressed on human endothelium and promotes angiogenesis by controlling directional migration [31]. A recent study reported the involvement of AMOT in EC migration through a key role in the mechanical signal’s integration detected by the migrating endothelium-mediated the contractility of actomyosin, essential for EC migration [24]. In accordance with this, we already reported that AMOT mediated the beneficial effects of sCD146 on EC migration [19] and daily sCD146 injection in gastrocnemius induced a better revascularization and tissue regeneration in a hindlimb ischemia mouse model [20]. Moreover, many studies showed that sCD146 was a biomarker of endothelial damage, explaining the positive correlation between its plasma level and disease progression in chronic kidney failure and diabetic nephropathy [32]. High plasma levels of sCD146 were also found in acute coronary syndrome and reflected the severity of pulmonary congestion better than the brain natriuretic peptide [22]. Even if sCD146 was involved in angiogenesis [33], the sCD146/AMOT signaling pathway has not been clearly explored. We previously demonstrated that AMOT overexpression was early, intensively and sustainably upregulated in post-ischemic situation in ischemic hindlimb mouse model. In this previous report, we observed an overexpression of the p80 isoform of AMOT, an isoform involved in new vessel formation and EC migration and a later decrease of p130 isoform involved in blood vessel maturation and stabilization [11].

Altogether, the involvement of sCD146 and AMOT in angiogenesis and tissue regeneration in post-ischemic situations and the high AMOT expression in neovascular endothelial cells during angiogenesis makes AMOT an ideal target for imaging of the myocardial infarction healing process.

Thus, we recently developed and validated an AMOT-targeting, PET imaging agent named ^68^Ga-sCD146, with an interesting early predictive value for delayed post-ischemic tissue recovery in hind limb ischemia model [11]. The preserved specificity for AMOT was confirmed by both ex vivo and in vivo experiments on the hind limb ischemia mouse model (respectively, by autoradiography on muscle sections and by microPET imaging with or without an excess of blocking peptide). In addition, a significant correlation between the intensity of ^68^Ga-sCD146 PET signal and AMOT expression quantified by immunohistochemistry, validated the quantification of AMOT using ^68^Ga-sCD146 PET imaging.

In this study, we applied ^68^Ga-sCD146 microPET imaging to evaluate and quantify AMOT expression at different times post-MI and compared it to ^68^Ga-RGD_2_, ^18^F-FDG microPET imaging, ultrasound imaging and histological Sirius-red staining analysis.

The most reported radiotracers at the clinical stage for imaging angiogenesis are based on the RGD motif targeting the αvβ3 integrin. ^68^Ga-RGD_2_ PET imaging is still under clinical evaluation in many cardiovascular and oncologic diseases, but might suffer from high background noise and a low specificity for angiogenesis of the targeted αvβ3 integrin itself [34,35,36]. Despite multiple clinical evaluations, the RGD-derived tracers are still under evaluation and further studies will be needed to establish the clinical utility of αvβ3 integrin targeting [37]. In our MI mouse model, ^68^Ga-sCD146 showed a significantly different microPET signal quantification on day 15 compared to sham mice, whereas ^68^Ga-RGD_2_ did not exhibit the least difference at this early time. In addition, analysis of ^68^Ga-sCD146 PET images showed a high hepatic tracer uptake. In the context of cardiac uptake evaluation, this observation did not preclude signal analysis. A significant hepatic uptake is also systematically described with one of the gold standard radiotracers for cardiac perfusion scintigraphy, ^99m^Tc-MIBI [38,39]. Chemical structure optimizations must now be considered in order to increase the hydrophilicity of ^68^Ga-sCD146, therefore improving pharmacokinetics via decreasing the hepatic retention, as already and successfully achieved with such as pegylated group addition or fluorinated RGD-derived radiotracers [40,41]. The superiority of ^68^Ga-sCD146 microPET signal was supported by significant correlations between ^68^Ga-sCD146 PET signal intensity at day 15 and delayed myocardial perfusion, myocardial recovery and lower myocardial fibrosis.

## 5. Conclusions

To conclude, this study demonstrated for the first time that AMOT was upregulated in post-MI mouse model and that AMOT expression could be considered as an informative key player in myocardial recovery processes. AMOT expression was successfully monitored by longitudinal ^68^Ga-sCD146 PET imaging and that constituted an early predictive value for delayed myocardial recovery. Additionally, this innovative PET imaging approach would be an asset in the evaluation of experimental therapeutic strategies aimed at promoting post-MI regenerative processes.

## Figures and Tables

**Figure 1 cells-10-02305-f001:**
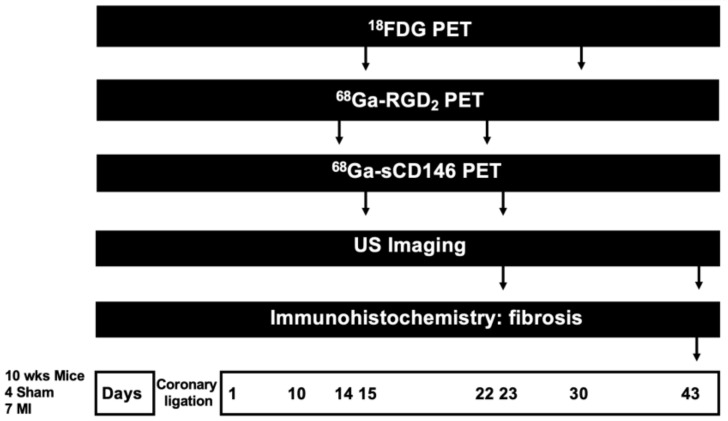
Experimental paradigm.

**Figure 2 cells-10-02305-f002:**
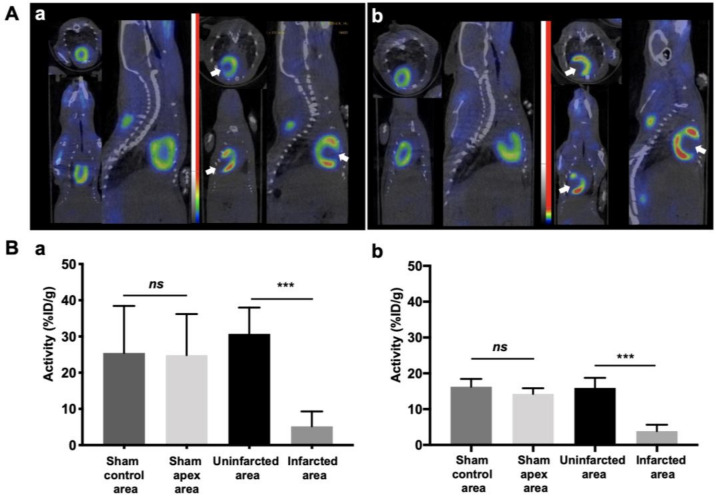
Myocardial infarction resulted in myocardial viability defect evaluated by ^18^F-FDG microPET imaging. (**A**) ^18^F-FDG microPET imaging showed a signal decrease in the infarcted myocardium identified by a white arrow at day 16 (**a**) and at day 30 (**b**) post-surgery. (**B**) Semi-quantitative analysis expressed as percentage of injected dose per gram of mouse (%ID/g mean ± SD) for day 16 (**a**) to day 30 (**b**) post-surgery showed a significant decreased in Apex area compared to control area (Mean_apex_ = 5.19 ± 4.10%ID/g, Mean_control_ = 30.7 ± 7.25%ID/g, *n* = 7; *** *p* = 0.0003 at day 16) and (Mean_apex_ = 3.84 ± 1.81%ID/g, Mean_control_ = 15.9 ± 2.81%ID/g, *n* = 7; *** *p* = 0.0003 at day 30). In contrast, uptake was homogeneous in sham hearts for both experiment time (Mean_apex_ = 24.8 ± 11.3%ID/g, Mean_control_ = 25.4 ± 12.9%ID/g, *n* = 4; *ns*: *p* = 0.443 at day 16) and (Mean_apex_, 14.3 ± 1.54%ID/g, Mean_control_ 16.2 ± 2.20%ID/g, *n* = 4; *ns*: *p* = 0.885 at day 30).

**Figure 3 cells-10-02305-f003:**
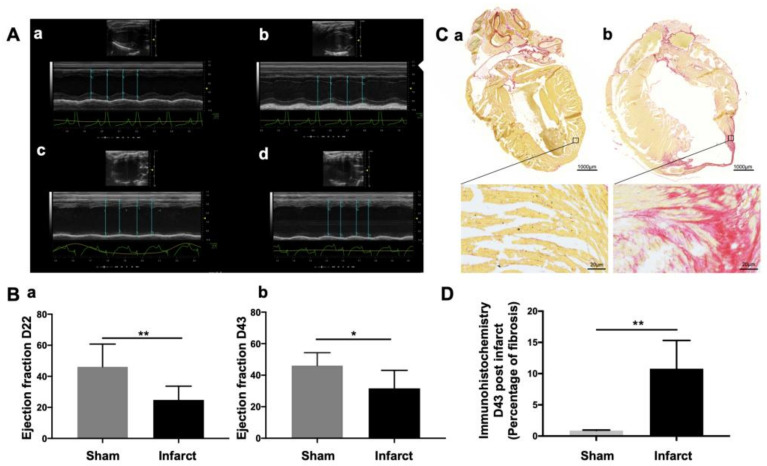
Ultrasound imaging and histological Sirius red staining analysis confirmed that myocardial infarction induced functional impairment. (**A**) Representative images of heart morphology and one-dimensional trace of the above line over time. (**B**) Quantitative analysis of ultrasound signal showed a significant decrease of the ventricular ejection fraction (VEF) compared to sham mice (**a**) (VEF_ischemic_ = 24.8 ± 8.84, *n* = 7, VEF_sham_ = 46.2 ± 14.5, *n* = 4; ** *p* = 0.009 at day 22) and (**b**) (VEF_ischemic_ = 31.7 ± 11.4, *n* = 7, VEF_sham_ = 46.2 ± 8.08, *n* = 4; * *p* = 0.04 at day 43). (**C**) Myocardial fibrosis following myocardial infarction. Histological Sirius red staining was performed 43 days post-surgery. Representative pictures of sham (**a**) and myocardial infarcted (**b**) heart sections showing fibrous tissue is stained red and myocardium is stained yellow. (**D**) Fibrosis quantification expressed as a percentage of fibrosis over the left ventricle showed a significant increased fibrosis in the infarcted heart compare to sham heart (Mean_ischemic_ = 10.79 ± 4.52, *n* = 7, Mean_sham_ = 0.88 ± 0.07, *n* = 4; ** *p* = 0.0061) at day 43 post-surgery.

**Figure 4 cells-10-02305-f004:**
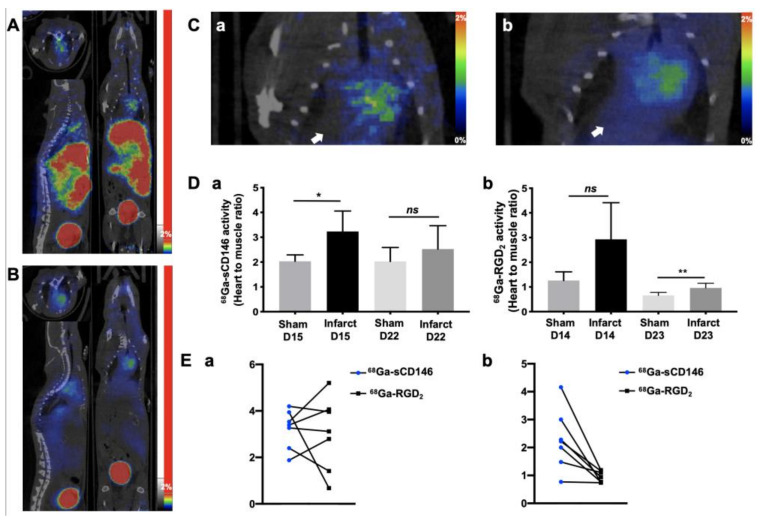
AMOT expression evaluated by ^68^Ga-sCD146 microPET imaging was early increased after myocardial infarction. (**A**) Representative whole body PET images of ^68^Ga-sCD146 on infarcted heart. (**B**) Representative whole body PET images of ^68^Ga-RGD_2_ on infarcted heart and PET imaging on infarcted heart. (**C**) Representative cardiac-focused images of ^68^Ga-sCD146 (**a**) and ^68^Ga-RGD_2_ (**b**) PET imaging on infarcted heart. White arrow indicates the infarcted myocardium area. (**D**) Semi-quantitative analysis of ^68^Ga-sCD146 PET uptake (**a**) expressed as heart-to-muscle ratio, showed an early and significant increase in infarcted hearts at day 15 compared to sham hearts (H/M_ischemic_ = 3.23 ± 0.82, *n* = 7, H/M_sham_ = 2.03 ± 0.25, *n* = 4; * *p* = 0.04) and a trend of increasing uptake at day 22 post-surgery (H/M_ischemic_ 2.52 ± 0.94, *n* = 7; H/M_sham_ = 2.02 ± 0.56, *n* = 4; *ns: p* = 0.30). Semi-quantitative analysis of ^68^Ga-RGD_2_ PET uptake (**b**), expressed as heart-to-muscle ratio, showed a delayed but significant increase in the infarcted heart compared to sham hearts at day 23 post-surgery (H/M_ischemic_ = 0.96 ± 0.18, *n* = 7, H/M_sham_ = 0.65 ± 0.12, *n* = 4; *p* = 0.02). No uptake modification was observed before this time (H/M_ischemic_= 2.92 ± 1.48; H/M_sham_ = 1.26 ± 0.13, *n* = 7, *ns: p* = 0.15 at day 14 post-myocardial infarction). (**E**) Semi-quantitative intra-animal comparison of both tracers, ^68^Ga-sCD146 and ^68^Ga-RGD_2_ at early times, respectively D14 and D15 after MI surgery (**a**). No significant difference was observed between ^68^Ga-sCD146 and ^68^Ga-RGD_2_ PET uptake in the infarcted heart at this time (n = 7; p = 0.93). Semi-quantitative intra-animal comparison was also realized a later time, respectively D22 and D23 for ^68^Ga-sCD146 and ^68^Ga-RGD_2_ (**b**). A significant difference was observed between ^68^Ga-sCD146 and ^68^Ga-RGD_2_ PET uptakes in the infarcted hearts at this later time (*n* = 7; ** *p* = 0.01).

**Figure 5 cells-10-02305-f005:**
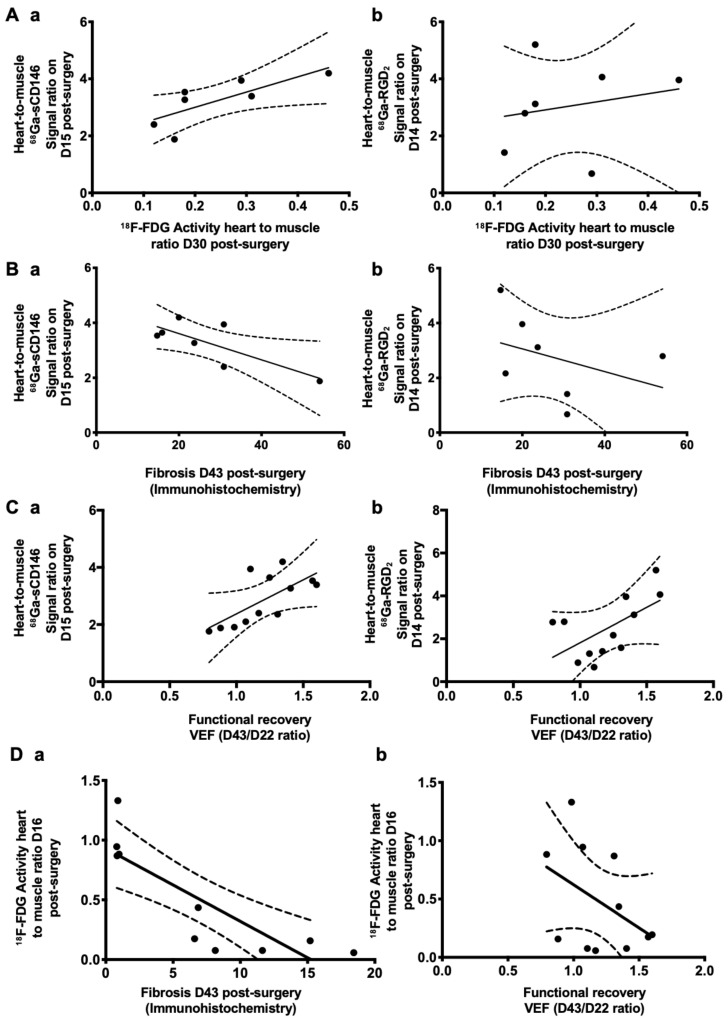
Correlation analysis. (**Aa**) Significant positive correlation between residual perfusion recovery perfusion recovery evaluated by ^18^F-FDG microPET signal expressed as heart-to-muscle ratio on day 30 and ^68^Ga-sCD146 microPET signal intensity expressed as heart-to-muscle ratio on day 15 post-myocardial infarction (Pearson R^2^ = 0.70; *p* = 0.04; *n* = 7). (**Ab**) No significant correlation was observed between individual residual perfusion recovery and ^68^Ga-RGD_2_ microPET signal intensity on day 14 post-ischemia (Pearson R^2^ = 0.21; *p* = 0.64; *n* = 7). (**Ba**): Significant negative correlation was detected between individual fibrosis quantification evaluated by histological Sirius red staining and expressed as percentage of fibrosis tissue on left ventricle 43 days post-myocardial infarction and ^68^Ga-sCD146 microPET signal intensity expressed as heart-to-muscle ratio on day 15 post-ischemia (Pearson R^2^ = −0.77; *p* = 0.04; *n* = 7). (**Bb**) No significant correlation was observed between individual fibrosis damages on day 43 post-ischemia and ^68^Ga-RGD_2_ PET signal on day 15 post-ischemia (Pearson R^2^ = −0.68; *p* = 0.06; *n* = 7). (**Ca**) a significant positive correlation was found between individual functional recovery explored with ultrasound imaging and expressed as the ratio between VEF 43 and 22 days post-myocardial infarction and ^68^Ga-sCD146 microPET signal intensity expressed as heart-to-muscle ratio on day 15 post-myocardial infarction (Pearson R^2^ = 0.68; *p* = 0.01; *n* = 12). (**Cb**) A significant correlation is identified between individual functional recovery and ^68^Ga-RGD_2_ microPET signal intensity on day 15 post-ischemia (Pearson R^2^ = 0.59; *p* = 0.04; *n* = 12). (**Da**) A significant negative correlation was identified between individual fibrosis on day 43 post-ischemia and ^18^F-FDG PET signal intensity ratio on day 16 post-ischemia (Pearson R^2^ = −0.83; *p* = 0.024; *n* = 10). (**Db**) No significant correlation was detected between functional recovery, ratio D43 on D22 post-myocardial infarction and ^18^F-FDG PET signal intensity ratio on day 16 post-ischemia (Pearson R^2^ = −0.39; *p* = 0.25; *n* = 10).

## Data Availability

Not applicable.

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
