# Peer review of "Comparison of a New 68Ga-Radiolabelled PET Imaging Agent sCD146 and RGD Peptide for In Vivo Evaluation of Angiogenesis in Mouse Model of Myocardial Infarction"

_cells, 2021, doi:10.3390/cells10092305_

Round 1
Reviewer 1 Report
In this preclinical study, the authors demonstrated that 68Ga-sCD146 could represent a promising PET radiotracer for tissue angiogenesis assessment in mycardial infarction.
The research methods are adequate and the results are very interesting.
There are some typo/grammar errors along the manuscript (including the abstract); please check and correct.
Author Response
In this preclinical study, the authors demonstrated that 68Ga-sCD146 could represent a promising PET radiotracer for tissue angiogenesis assessment in mycardial infarction.
The research methods are adequate and the results are very interesting.
There are some typo/grammar errors along the manuscript (including the abstract); please check and correct.
We thank the reviewer for his encouraging comment and the manuscript has been proofread for typos/grammar according to his demand.
Reviewer 2 Report
The work of Moyon and co-authors try to elucidate the potential of the new tracer, 68Ga-sCD146, as tool to assess heart functional receovery after myocardial infarction.
The work is well planned, the experiments well performed.
I have some concerns:
1- Proof read the english, some sentences present repetitions.
2- In the text immunofluorescence staining is reported, but no results about tissue section from euthanized succinate-treated mice are reported. Please provide this data, which definitely help to support your conclusions made on the PET images.
3- Please check for misspelling. Pag. 6, lane 228 and Pag. 7, lane 272, you report TEP instead of PET (I suppose).
4- Pag. 7, lane 278: check for repetition.
5- Could you please provide axial projection images of these mice in Figure 4? It will help in comparing these images with those obtained with 18F-FDG.
6- In addition, it would make more easy for the reader to conclude about the better performance of the new tracer, if you match the PET images of 68Ga-sCD146 with 18F-FDG!
7- In Figure 4, PET signal at different level in cest area are presented, which do not allow a clear and inequivocable comparison between 68Ga-sCD146 and 68Ga-RGD2. Why an intra-animal comparison of both tracers has not been carried out? This could have shown better the superiority of the proposed tracer.
Author Response
The work of Moyon and co-authors try to elucidate the potential of the new tracer, 68Ga-sCD146, as tool to assess heart functional receovery after myocardial infarction.
The work is well planned, the experiments well performed.
We thank the reviewer for his encouraging comment.
I have some concerns:
- Proof read the english, some sentences present repetitions.
The manuscript has been proofread for typos/grammar according to the reviewer’s demand.
- In the text immunofluorescence staining is reported, but no results about tissue section from euthanized succinate-treated mice are reported. Please provide this data, which definitely help to support your conclusions made on the PET images.
This paragraph was removed as no immunofluorescence was carried out in this project.
- Please check for misspelling. Pag. 6, lane 228 and Pag. 7, lane 272, you report TEP instead of PET (I suppose).
These typos were rectified.
- 7, lane 278: check for repetition.
The sentence was reformulated.
- Could you please provide axial projection images of these mice in Figure 4? It will help in comparing these images with those obtained with 18F-FDG.
The figure 4 has been modified according to the reviewer’s demand.
- In addition, it would make more easy for the reader to conclude about the better performance of the new tracer, if you match the PET images of 68Ga-sCD146 with 18F-FDG!
Given the fact that 18F-FDG and 68Ga-sCD146 PET were not acquired at the same time, it was unfortunately technically impossible to match both PET tracers images. The main limit is the orientation of the animal's heart is not exactly the same from one imaging session to another.
- In Figure 4, PET signal at different level in cest area are presented, which do not allow a clear and inequivocable comparison between 68Ga-sCD146 and 68Ga-RGD2. Why an intra-animal comparison of both tracers has not been carried out? This could have shown better the superiority of the proposed tracer.
As the reviewer commented, an intra-animal comparison of both tracers could have shown better the superiority of our radiotracer. To meet this demand, we modified figure 4 and added a comparison of both tracers for each animal. These results allow to highlight that the real impact of this new tracer is the predictive value of a better perfusion and myocardial recovery described by the various correlation studies and which wasn’t found with 68Ga-RGD2.
Reviewer 3 Report
The authors presented the development of a new PET imaging strategy for angiogenesis. There are a few points that the authors can consider for possible improvement of their manuscript.
- In the Introduction, there is a statement "Indeed, PET imaging targeting integrins or 53 molecules of the VEGF pathway did not succeed to date and...". Please provide references for this, and please also provide a brief explanation on why they did not work.
- In the study design (Figure 1), please briefly explain/justify how the days (14 days for early and 22 days for later) were selected.
- In comparison between sCD146 and RGD (Figure 4), why only small portion of the body was display? why not display the whole body as in Figure 2 (FDG imaging)? Also, please put arrows to point to the infarction site in (A). For (B)b, which is quantification for RDG, it looks like a significant difference between Shan and Infarct on Day 14, but P=0.15 is not significant?!?!
- For mice imaged, what were the serum level of sCD146? If the circulating sCD146 is higher, there will be some competition for the binding of [Ga-68]-sCD146 to AMOT??
- All the figure legends need to be more detailed to separate a and b within (A) and (B).
Author Response
The authors presented the development of a new PET imaging strategy for angiogenesis. There are a few points that the authors can consider for possible improvement of their manuscript.
- In the Introduction, there is a statement "Indeed, PET imaging targeting integrins or 53 molecules of the VEGF pathway did not succeed to date and...". Please provide references for this, and please also provide a brief explanation on why they did not work.
The introduction was updated according to the reviewer’s demand and a paragraph was added to clarify the message.
- In the study design (Figure 1), please briefly explain/justify how the days (14 days for early and 22 days for later) were selected.
Due to the cardiovascular weakness of the MI model, we had to limit the number of anesthesia and thus perform successive evaluations as early as possible for the three tracers on the same animals. The clinical condition allowed us to start the evaluation not before D14. The kinetics of AMOT expression not being defined in mouse myocardial infarction model, we evaluated the expression of AMOT as soon as possible and then at a later time showing a decreasing expression.
- In comparison between sCD146 and RGD (Figure 4), why only small portion of the body was display? why not display the whole body as in Figure 2 (FDG imaging)?
Figure 4 was updated according to the reviewer’s demand. We have added a paragraph mentioning the pharmacokinetics of this new tracer in the discussion.
Also, please put arrows to point to the infarction site in (A).
Figure 4 was updated accordingly.
For (B)b, which is quantification for RDG, it looks like a significant difference between Shan and Infarct on Day 14, but P=0.15 is not significant?!?!
As the reviewer commented and we totaly agree, “it looks like a significant difference between Sham and Infarct on Day 14”. Yet, we can confirm that the statistical analysis did not result in a significant difference. This is probably due the fact that despite the strong PET signal observed with the 68Ga-RGD2, there is a really high inter-individual variation in signal intensity that impede to get significant difference.
- For mice imaged, what were the serum level of sCD146? If the circulating sCD146 is higher, there will be some competition for the binding of [Ga-68]-sCD146 to AMOT??
Even if the level of interest of sCD146 was not evaluated in these mice, experiments carried out previously on the same MI mice models showed a circulating sCD146 serum level of approximately 15ng/mL. Even if we cannot exclude a competition on AMOT, the serum concentrations of sCD146 found in this model remain 100 times lower than the quantity of tracers injected to the mouse. The large excess of injected tracer compared to the circulating sCD146 seems to allow the competition to be neglected. The presence of sCD146 circulating in this mouse model enables comparison with the pathological situation in humans. Indeed, it has been shown that serum sCD146 levels are higher in patients with MI (approximatively 300ng/mL) compared to control patients. The scaled injection of a large excess of our tracer in humans will also undergo limited competition with the sCD146 present in this pathology [6] [7].
- All the figure legends need to be more detailed to separate a and b within (A) and (B).
Figure legends were updated accordingly.
Round 2
Reviewer 3 Report
The authors addressed most of the issues raised by this reviewer.
From Fig. 4, comparison between sCD146 (A) and RGD2 (B) showed that the RGD2 ligand resulted in a clean image while sCD146 had a very high biliary clearance as well as high renal uptake. Perhaps, sCD146 needs some modification. Please discuss this in Discussion. For (C), what are the white arrows pointing to? Please clarify in the figure legend.
Author Response
From Fig. 4, comparison between sCD146 (A) and RGD2 (B) showed that the RGD2 ligand resulted in a clean image while sCD146 had a very high biliary clearance as well as high renal uptake. Perhaps, sCD146 needs some modification. Please discuss this in Discussion. For (C), what are the white arrows pointing to? Please clarify in the figure legend.
As the reviewer commented and we totaly agree, the discussion and figure 4 legend have been modified. Changes are underlined in green in the manuscript